# GRP75 Modulates Endoplasmic Reticulum–Mitochondria Coupling and Accelerates Ca^2+^-Dependent Endothelial Cell Apoptosis in Diabetic Retinopathy

**DOI:** 10.3390/biom12121778

**Published:** 2022-11-29

**Authors:** Yan Li, Hong-Ying Li, Jun Shao, Lingpeng Zhu, Tian-Hua Xie, Jiping Cai, Wenjuan Wang, Meng-Xia Cai, Zi-Li Wang, Yong Yao, Ting-Ting Wei

**Affiliations:** 1Department of Ophthalmology, The Affiliated Wuxi People’s Hospital of Nanjing Medical University, Wuxi 214023, China; 2Center of Clinical Research, The Affiliated Wuxi People’s Hospital of Nanjing Medical University, Wuxi 214023, China

**Keywords:** diabetic retinopathy, ER–mitochondria coupling, IP3R1–GRP75–VDAC1 axis, mitochondrial Ca^2+^, apoptosis

## Abstract

Endoplasmic reticulum (ER) and mitochondrial dysfunction play fundamental roles in the pathogenesis of diabetic retinopathy (DR). However, the interrelationship between the ER and mitochondria are poorly understood in DR. Here, we established high glucose (HG) or advanced glycosylation end products (AGE)-induced human retinal vascular endothelial cell (RMEC) models in vitro, as well as a streptozotocin (STZ)-induced DR rat model in vivo. Our data demonstrated that there was increased ER–mitochondria coupling in the RMECs, which was accompanied by elevated mitochondrial calcium ions (Ca^2+^) and mitochondrial dysfunction under HG or AGE incubation. Mechanistically, ER–mitochondria coupling was increased through activation of the IP3R1–GRP75–VDAC1 axis, which transferred Ca^2+^ from the ER to the mitochondria. Elevated mitochondrial Ca^2+^ led to an increase in mitochondrial ROS and a decline in mitochondrial membrane potential. These events resulted in the elevation of mitochondrial permeability and induced the release of cytochrome c from the mitochondria into the cytoplasm, which further activated caspase-3 and promoted apoptosis. The above phenomenon was also observed in tunicamycin (TUN, ER stress inducer)-treated cells. Meanwhile, BAPTA-AM (calcium chelator) rescued mitochondrial dysfunction and apoptosis in DR, which further confirmed of our suspicions. In addition, 4-phenylbutyric acid (4-PBA), an ER stress inhibitor, was shown to reverse retinal dysfunction in STZ-induced DR rats in vivo. Taken together, our findings demonstrated that DR fueled the formation of ER–mitochondria coupling via the IP3R1–GRP75–VDAC1 axis and accelerated Ca^2+^-dependent cell apoptosis. Our results demonstrated that inhibition of ER–mitochondrial coupling, including inhibition of GRP75 or Ca^2+^ overload, may be a potential therapeutic target in DR.

## 1. Introduction

Diabetic retinopathy (DR) is a major microvascular complication of diabetes mellitus (DM) that leads to sight impairment in one-third of diabetic patients [1]. In the retina, DR causes vascular endothelial dysfunction, pericyte loss, and neuronal abnormalities through multiple interlinked pathways [2,3]. Since retinal microvascular endothelial cells (RMECs) line the inner wall of vessels, its dysfunction is an early event in DR due to long-term contact with glucose-rich blood [4,5]. Impairment of the RMECs leads to an increase in endothelial permeability, which eventually fuels vascular complications and damage of the blood–retinal barrier (BRB) [6]. Thus, improving RMEC dysfunction is critical for remission of DR-induced vision loss.

Mitochondria are central organelles for energy production and are closely related to cell fate. The mitochondria regulate cellular metabolism and redox homeostasis, which mediate pathological processes in many diseases, including DR [7]. Furthermore, the mitochondria play an important role in calcium ion (Ca^2+^) homeostasis. Imbalance of mitochondrial Ca^2+^ homeostasis has been shown to be associated with a variety of diseases including diabetes [8]. Under normal physiological conditions, the mitochondria do not harbor Ca^2+^ [9]. However, since the mitochondria have good Ca^2+^ buffering capacity, a large amount of Ca^2+^ enter mitochondria under cellular stress, resulting in the surge of mitochondrial Ca^2+^ [10]. The rate of energy metabolism regulates mitochondrial-free Ca^2+^ that stimulate mitochondrial Ca^2+^-sensitive enzymes [11]. In addition, the balance between formation and scavenging of free radicals was shown to be disrupted by excess mitochondrial Ca^2+^, which exacerbates oxidative stress and induces apoptosis [12]. The endoplasmic reticulum (ER) is the main intracellular organelle that stores Ca^2+^, which is a crucial factor for cellular homeostasis. In addition, the ER is a site of protein synthesis, folding, and modification. Previous data have demonstrated that a high concentration of Ca^2+^ in the ER lumen ensures proper functioning of the ER [13,14]. Dysregulation of ER Ca^2+^ can trigger an ER stress (ERS) response that leads to cell death [15], and sustained outflow of ER Ca^2+^ results in elevation of cytoplasmic Ca^2+^. The elevation of Ca^2+^ initiates a series of physiological reactions such as caspase activation, release of apoptotic factors from the mitochondria, and direct activation of Ca^2+^-dependent endonucleases that cleave cellular DNA [16,17]. Therefore, intracellular Ca^2+^ imbalance defines the initiation of DR, and thus adjustment of the mitochondrial and/or ER dysfunction may ameliorate the microvascular endothelial injury.

Maintenance of cellular homeostasis is related to the biological functions of several organelles that play coordinated or antagonistic roles. Crosstalk between organelles, especially the interaction between the ER and mitochondria, has attracted a large amount of research interest in recent decades [18]. Mitochondria-associated membranes (MAMs) tether the mitochondria to the ER and regulate cell processes through a cascade of signaling events such as Ca^2+^ homeostasis, oxidative stress, apoptosis, and lipid biosynthesis [19,20]. Moreover, there are many core proteins in mammals that either directly regulate the physical connectivity of the MAMs or indirectly regulate the binding complexes in the MAMs. Bridging the inositol triphosphate receptor 1 (IP3R1)–glucose-regulated protein 75 (GRP75)–voltage-dependent anion channel 1 (VDAC1) complex, which is one of the main members of the core proteins, regulates the mitochondria Ca^2+^ homeostasis [21]. GRP75 interconnects two organelles by assembling the IP3R1–GRP75–VDAC1 complex, which enhances the ER–mitochondria interaction. Therefore, increased GRP75 expression accelerates Ca^2+^ transfer from the ER to the mitochondria, causing Ca^2+^ overload, which prompts mitochondrial membrane potential impairment and increased mitochondria ROS production [22]. In contrast, decreased expression of GRP75 suppresses mitochondrial Ca^2+^ levels [23]. Together, this evidence indicates that any abnormalities in the IP3R1–GRP75–VDAC1 axis induces mitochondrial Ca^2+^ overload and apoptosis.

In this study, we aimed to evaluate the role of the IP3R1–GRP75–VDAC1 axis in modulating Ca^2+^ homeostasis in RMECs. Our results revealed that ERS transports Ca^2+^ to the mitochondria via the IP3R1–GRP75–VDAC1 axis and promotes the formation of MAMs mainly by increasing the GRP75 expression, which ultimately induces mitochondrial Ca^2+^ overload leading to apoptosis. Thus, this pathway presents a potential treatment target for DR.

## 2. Materials and Methods

### 2.1. Materials and Reagents

Glucose, streptozocin (STZ), and Evans blue were purchased from Sigma-Aldrich (St. Louis, MO, USA), while advanced glycosylation end products (AGEs) were acquired from BioVision (Palo Alto, CA, USA). Tunicamycin (TUN) was obtained from Abcam (Cambridge, CA, USA), while 4-phenylbutyric acid (4-PBA), BAPTA-AM, and Protein A/G magnetic beads were purchased from MedChemExpress (Lowell, NJ, USA). Cell counting kit-8 (CCK8), JC-1 assay kit, Calcein/PI assay kit, MPTP assay kit, Mito-Tracker Green, ER-Tracker Red, and a mitochondria isolation kit were purchased from Beyotime (Nantong, China). On the other hand, the TUNEL assay kits were purchased from Roche (Basel, Switzerland). MitoSOX™ Red, DAPI, ECL kits, and goat anti-mouse/rabbit IgG (H+L, Alexa Fluor 555/488) were acquired from Thermo Fisher (Waltham, CA, USA). We used primary antibodies including GRP75, Cyt c, Bcl-xl, Bax, cleaved caspase-3, and CHOP (Cell Signaling Technology, Boston, MA, USA); 4-HNE and Brn3a (Abcam, Cambridge, CA, USA); IP3R1, VDAC1, VEGF, 8-OHDG, and 3-NT (Santa Cruz Biotechnology, Santa Cruz, CA, USA); and COX IV, Bcl-2, and β-actin (Proteintech, Wuhan, China). More detailed information about the materials and reagents is offered in Appendix A.

### 2.2. Animals and Models

Male Sprague-Dawley rats (SD, 8 weeks old, 200–250 g) were purchased from Changzhou Cavans Experimental Animal Co., Ltd. (Changzhou, China). The animals were housed at 22–26 °C, while food and water were provided ad libitum during the entire experiment. All experimental procedures were performed according to the National Institutes of Health Guidelines for the Care and Use of Laboratory Animals and were approved by the Animal Care and Use Committee of Nanjing Medical University (SYXK(SU)2020-0010).

The rats were divided into 2 groups randomly: normal control rats (Ctrl, *n* = 15) and diabetes rats (*n* = 45). Diabetes was induced by intraperitoneal injection of STZ (1.5%, 60 mg/kg) dissolved in citrate buffer (pH 4.5, 0.1 mol/L), while the control group received an equivalent volume of citrate buffer. Fasting blood glucose level was assessed on the third day after STZ treatment. Only 32 rats with diabetes were modelled successfully, for which random blood glucose was higher than 16.7 mM. The 32 rats were once again randomized into 2 groups: the diabetes group (*n* = 16) and the diabetes with 4-PBA treatment group (STZ + 4-PBA, *n* = 16). 4-PBA (500 mg/kg) was administered daily by oral gavage from week 4 to week 14. The diabetes group was treated with equal volumes of the vehicle solution. The rats were sacrificed by overdose of anesthesia at the 14th week, and then the eyes and retinas were isolated for further analyses.

### 2.3. Cell Culture

Human retinal microvascular endothelial cells (RMECs) were acquired from the Beina Chuanglian Biotechnology Institute (Beijing, China). The cells were maintained in complete DMEM medium (Gibco, Grand Island, New York, NY, USA) with 10% FBS (Gibco Grand Island, New York, NY, USA) and 1% penicillin and streptomycin (Beyotime, Nantong, China) at 5% CO_2_, 37 °C.

### 2.4. siRNA Transfection

The RMECs were transfected with GRP75 or negative control (NC) siRNA at 80% density with a riboFECT CP Transfection Kit (RiboBio, Guangzhou, China). The RMECs were cultivated in 6-well plates overnight. A total of 120 μL Opti-MEM medium including 12 μL riboFECT CP, 100 nM GRP75 siRNA, or 100 nM NC was blended in a 6-well plate. Detailed siRNA information is shown in Appendix A.

### 2.5. Cell Counting Kit 8 (CCK-8)

We analyzed cell viability using the CCK8 kit (Beyotime, Nantong, China). RMECs (1000 per well) were planted in 96-well plates followed by the addition of CCK-8 solution (100 μL/each well). The cells were incubated for 1.5 h at 37 °C in a 5% CO_2_ incubator. Absorbance was detected at 450 nm by a microplate reader.

### 2.6. ER and Mitochondrion Staining

The ER was stained using the ER-Tracker Red kit (1 μM, Red, Beyotime, Nantong, China). The mitochondria were stained using MitoTracker-Green (200 nM, Green, Beyotime, Nantong, China) or MitoTracker-Red (200 nM, Red, Beyotime, Nantong, China). The cells were assessed by laser scanning confocal microscopy (40×, Leica, Weztlar, Germany).

### 2.7. Detection of Mitochondrial Ca^2+^

Mitochondrial Ca^2+^ was detected using Rhod-2 AM (2 μM, Red, YEASEN, Shanghai, China), which contains Pluronic F-127 (0.02%, Beyotime, Nantong, China). In addition, MitoTracker-Green (200 nM, Green, Beyotime, Nantong, China) was used to co-stain the mitochondria. The cells were evaluated under the laser scanning confocal microscope (40×, Leica, Weztlar, Germany).

### 2.8. Analysis of Mitochondrial ROS and Mitochondrial Membrane Potential (MMP)

The levels of mitochondrial ROS were assayed by MitoSOX™ Red (Red, Thermo Fisher, Waltham, MA, USA). Briefly, the RMECs were stained with the MitoSOX™ Red (5 μM, 10 min) and then co-stained with MitoTracker-Green (200 nM, Green, 30 min, Beyotime, Nantong, China). The mitochondrial ROS was analyzed by laser scanning confocal microscopy (40×, Leica, Weztlar, Germany).

On the other hand, MMP was assessed by a JC-1 assay kit (Beyotime, Nantong, China). Briefly, the RMEC cells were incubated with JC-1 staining solution. The cells were visualized under the laser scanning confocal microscope (40×, Leica, Weztlar, Germany). MMP was calculated as the ratio of red/green using ImageJ software.

### 2.9. Measurement of Mitochondrial Permeability Transition Pore (mPTP)

The mitochondrial permeability transition pore opening was analyzed by an mPTP assay kit (Beyotime, Nantong, China). Calcein-AM dye selectively gathers inside the mitochondria and stimulates green fluorescence, while cobalt chloride (CoCl_2_) quenches the green fluorescence of calcein in the cytoplasm. Once mPTP is opened, calcein is released from the mitochondria and then quenched by CoCl_2_. Therefore, the stronger the calcein green fluorescence, the less mPTP opening. Briefly, the RMECs were loaded with Calcein-AM dye (2 μM) containing CoCl_2_ (250 μM). The Mitotraker-red (200 nM, 30 min, Beyotime, Nantong, China) was then used to co-stain the mitochondria. The cells were analyzed by laser scanning confocal microscopy (40×, Leica, Weztlar, Germany).

### 2.10. Calcein-AM/Propidium Iodide (PI) Staining

Live or dead cells were analyzed by a Calcein-AM/PI double stain kit (Beyotime, Nantong, China). The RMECs were stained with Calcein-AM (2 μM) and PI (5 μM), and then the analysis of the stained cells was immediately performed by laser scanning confocal microscopy (20×, Leica, Weztlar, Germany).

### 2.11. Vascular Permeability of the Blood–Retinal Barrier (BRB)

Vascular permeability was evaluated by assessing Evans blue dye leakage from retinal vessels as previously described [24]. The rats received intravenous injection (through the tail vein) of Evans blue (3%, dissolved in 0.9% NaCl). After 2 h, the animals turned blue. Thereafter, the eyes were isolated and the retinas were removed from the eyecup, and following this, we used the BX-51 light microscope (10×, Olympus, Tokyo, Japan) for observation. For quantification, retinas were weighed and then incubated in 150 μL formamide for 18 h at 78 °C to extract the Evans blue. The extract was centrifuged at 12,000× *g*, 30 min at 4 °C, and the absorbance was determined at 620 nm. The results were expressed in microgram per gram of retina.

### 2.12. Retinal Digestion and Periodic Acid–Schiff (PAS) Staining

Retinal PAS staining is a method used to detect the amounts of acellular capillaries in the retina. The retinas were fixed in 4% paraformaldehyde for 1.5 h and digested in 3% trypsin for 2 h at 37 °C. The samples were gently washed until the retinal leaves were completely spread. Visualized vessels were dried on carrier slides and then retinal capillaries were dyed with the PAS/haematoxylin stain kit (Solarbio, Beijing, China), following the manufacturer’s instructions. The samples were assessed under a light microscope (20×, Olympus, Tokyo, Japan).

### 2.13. HE Staining

To explore morphological alterations in the retina, we performed hematoxylin and eosin (H&E) staining. Briefly, eye paraffin-embedded tissues were cut into 5 μm sections at the periphery of the retinas. The slides were deparaffinized, rehydrated, and then stained using the H&E staining kit (BOSTER, Wuhan, China). The samples were analyzed using a light microscope (20×, Olympus, Tokyo, Japan).

### 2.14. TUNEL Assay

Apoptosis was evaluated using the TUNEL assay kit (Roche, Basel, Switzerland). Briefly, retinal frozen sections were permeabilized with 0.5% TritonX-100 and then incubated with TUNEL reaction mixture. The samples were then counterstained with DAPI (1:1000). The analysis of the samples was performed using laser scanning confocal microscopy (20×, Leica, Weztlar, Germany).

### 2.15. Immunohistochemistry and Immunofluorescence

Paraffin-embedded tissues were cut into 5 μM sections. The sections were stained with an immunohistochemistry kit (BOSTER, China). Briefly, the sections were incubated with 3% H_2_O_2_ for 15 min, and then antigen repair was performed in citrate buffer for 20 min. After blocking with 5% BSA for 1 h, the samples were incubated with 4-HNE (1:200, Abcam, Cambridge, CA, USA), Brn3a (1:500, Abcam, Cambridge, CA, USA), 8-OHDG (1:100, Santa Cruz Biotechnology, Santa Cruz, CA, USA), VEGF (1:100, Santa Cruz Biotechnology, Santa Cruz, CA, USA), or 3-NT (1:100, Santa Cruz Biotechnology, Santa Cruz, CA, USA) overnight at 4 °C. Thereafter, the samples were incubated with polymerized HRP-labeled anti-rabbit/mouse IgG and then detected with diaminobenzidine (DAB, BOSTER, Wuhan, China). Finally, the sections were counterstained with hematoxylin, then observed with a light microscope (20×, Olympus, Tokyo, Japan).

Cells or frozen sections were fixed with 4% paraformaldehyde for 15 min, then permeabilized with 0.2% Triton X100 (Beyotime, Nantong, China) for 15 min and blocked with 5% BSA for 1 h. Thereafter, the samples were incubated with CHOP (1:500, CST, Boston, MA, USA), c-caspase-3 (1:400, CST, Boston, MA, USA), or Rhod (1:100, Santa Cruz Biotechnology, Santa Cruz, CA, USA) overnight at 4 °C. The samples were then incubated with goat anti-rabbit/mouse IgG (H+L) and Alexa Fluor 555/488 (Thermo Fisher, Waltham, MA, USA) while nuclei were stained with DAPI (1:1000). Fluorescence was detected using laser scanning confocal microscopy (20×/40×, Leica, Weztlar, Germany), and we quantified the fluorescence intensity using ImageJ software (ImageJ, Bethesda, AR, USA).

### 2.16. Co-Immunoprecipitation

Co-immunoprecipitations (Co-IP) was performed as previously with minor adaptations [25]. The RMECs were lysed for 30 min using Pierce™ IP lysis buffer (Thermo Fisher, Waltham, MA, USA) containing protease inhibitor (Roche, Basel, Switzerland). The lysed cells were centrifuged at 12,000× *g* for 15 min at 4 °C to obtain the supernatant, and then protein concentration was determined using the BCA, assay kit (Beyotime, Nantong, China). A total of 1/10 of the supernatant was kept as the input group and the remaining 500 μL was taken for immunoprecipitation. Here, the supernatant was incubated with 1 μg primary antibody, IP3R1 or VDAC1, or normal control IgG overnight at 4 °C. The samples were incubated with 50 μL protein A/G magnetic beads (MedChemExpress, South Brunswick, NJ, USA) at 4 °C for 3 h and then centrifuged at 4 °C for 5 min at 1000× *g*. The magnetic beads were washed five times with buffer, and then proteins were eluted from the beads. The samples were boiled for 5 min in 1 × loading buffer and then resolved in SDS-PAGE. The samples were further analyzed by Western blot.

### 2.17. Western Blot Analysis

RMECs and tissues were extracted using RIPA lysis buffer (Beyotime, Nantong, China). Protein concentrations were quantified using the BCA, kit (Beyotime, Nantong, China). The samples were resolved in an electrophoresis set up and then transferred to PVDF membranes. After blocking with 5% milk for 2 h, the membranes were incubated with the following primary antibodies: GRP75 (1:1000, CST, Boston, MA, USA), IP3R1 (1:200, Santa Cruz Biotechnology, Santa Cruz, CA, USA), VDAC1 (1:200, Santa Cruz Biotechnology, Santa Cruz, CA, USA), Cyt c (1:1000, CST, Boston, MA, USA), Bcl-2 (1:2000, Proteintech, Wuhan, China), Bcl-xl (1:1000, CST, Boston, MA, USA), Bax (1:1000, CST, Boston, MA, USA), c-caspase-3 (1:1000, CST, Boston, MA, USA), COX IV (1:5000, Proteintech, Wuhan, China), or β-actin (1:5000, Proteintech, Wuhan, China) at 4 °C overnight. Thereafter, the blots were incubated with horseradish peroxidase-conjugated secondary antibody for 2 h and then analyzed using the chemiluminescence kit (Thermo Fisher, Waltham, MA, USA).

### 2.18. Statistical Analysis

At least three rounds of each experiment were performed. All data are expressed as mean ± SD (standard deviation). Statistical analyses were performed using Student’s two-tailed *t*-test or one-way ANOVA with Tukey’s honest significant difference test using GraphPad Prism version 8.0 (GraphPad Software, San Diego, CA, USA). A *p* < 0.05 was considered to be statistically significant.

## 3. Results

### 3.1. ER–Mitochondria Coupling Was Increased via the IP3R1–GRP75–VDAC1 Axis under DR

Numerous studies have shown that ER and mitochondrial dysfunction are involved in the development of DR [26,27]. Additional studies will be required to underline the relationship between ER and mitochondria. In this study, we used HG or AGE-induced RMECs to model DR conditions in vitro. ER-Tracker Red (1 μM), Mito-Tracker Green (200 nM), and Rhod-2 AM (2 μM) were used as ER, mitochondrial, and mitochondrial Ca^2+^ markers, respectively. Our data showed that ER enhanced the binding to mitochondria in RMECs under HG or AGE incubation (Figure 1A), a phenomenon that was accompanied by elevated mitochondrial Ca^2+^ levels (Figure 1B). ER is an essential intracellular Ca^2+^ storage pool, and Ca^2+^ is released from the ER on cellular stimulation. As previously mentioned, the IP3R1–GRP75–VDAC1 complex regulates the ER–mitochondria coupling and Ca^2+^ signaling [28]. Given that GRP75 plays a bridging role in the complex, we employed co-immunoprecipation (co-IP) to examine the interactions amongst IP3R1, GRP75, and VDAC1. Our findings demonstrated that under HG or AGE conditions, GRP75 is pulled down by IP3R1, and VDAC1 was pulled down by immunoprecipitation using an anti-GRP75 antibody (Figure 1C,D). The above results showed that increase in the ER–mitochondria coupling under DR conditions may be mediated by the IP3R1–GRP75–VDAC1 axis.

### 3.2. Inhibiting GRP75 Rescued ER–Mitochondria Coupling under DR Conditions

To further investigate the role of the IP3R1–GRP75–VDAC1 complex in the ER–mitochondrial coupling, RMECs were transfected with GRP75-specific siRNA, and then the knockdown efficiency was verified by Western blot analysis (Figure 2A). The analysis demonstrated suppression of the ER–mitochondrial coupling after knockdown of GRP75 under HG or AGE incubation (Figure 2B). Consequently, the knockdown of GRP75 was shown to significantly reduce mitochondria Ca^2+^ in HG or AGE-induced RMECs (Figure 2C). These results robustly demonstrated that the IP3R1–GRP75–VDAC1 axis regulates ER–mitochondria coupling and Ca^2+^ signaling under DR conditions.

We also analyzed whether the increased ER–mitochondria coupling could affect mitochondrial function by assessing the mitochondrial membrane potential (ΔΨm) and mitochondrial ROS. JC-1 and Mito-SOX are ideal fluorescent probes for the detection of mitochondrial membrane potential (ΔΨm) and mitochondrial ROS, respectively. Interestingly, the knockdown of GRP75 was shown to reverses the decline of mitochondrial membrane potential and the increase in mitochondrial ROS caused by HG (Figure 2D,E). Similar results were observed in the AGE treatment group (Appendix A). However, the mechanism of ER–mitochondrial-coupling-induced mitochondrial dysfunction remains unclear.

### 3.3. Mitochondria Ca^2+^-Induced Mitochondrial Dysfunction and Apoptosis

Our findings demonstrated that ER–mitochondria coupling transferred Ca^2+^ into the mitochondria through the IP3R1–GRP75–VDAC1 axis. The imbalance of intracellular Ca^2+^ in the retina defined retinal microvascular dysfunction in DR [29]. We evaluated whether mitochondrial dysfunction can be induced by increased mitochondrial Ca^2+^. To validate this hypothesis, we used BAPTA-AM, an intracellular calcium chelator, for analysis. The data showed that BAPTA-AM significantly blocked the production of mitochondrial ROS after HG or AGE treatment (Figure 3A). In addition, BAPTA-AM reversed the decrease in mitochondrial membrane potential caused by HG or AGEs (Figure 3B). These results indicated that elevation of Ca^2+^ in the mitochondria affects mitochondrial dysfunction. Decreased cell viability is associated with mitochondrial dysfunction. As shown in Figure 3C, live cells were stained with green fluorescent Calcein-AM, while the dead cells were stained with red fluorescent PI. In sync with the CCK8 assay results, BAPTA-AM decreased the percentage of dead cells caused by HG or AGEs (Figure 3D). However, the mechanism of mitochondrial Ca^2+^-induced damage in RMECs remains undefined. Decreased mitochondrial membrane potential and increased mitochondrial ROS is a hallmark in the early stages of apoptosis. Here, we analyzed apoptosis-related proteins, including anti-apoptotic-related proteins (Bcl-2 and Bcl-xl) and pro-apoptosis-related proteins (Bax and cleaved caspase-3). Western blot analysis showed that BAPTA-AM upregulated anti-apoptotic proteins but downregulated apoptosis-related proteins in HG or AGE-treated cells (Figure 4A), which was consistent with the results of the immunofluorescence of cleaved caspase-3 (Figure 4B). 

To further investigate the mechanisms of mitochondrial Ca^2+^-induced activation of caspase-3 and apoptosis, we evaluated mitochondrial/cytoplasmic cytochrome C (Cyt c), an inducer of caspase-3 activation, under HG or AGE conditions. We showed that HG or AGEs induced increased levels of cytoplasmic Cyt c and decreased levels of mitochondrial Cyt c. The cytoplasmic Cyt c further induced caspase 3 activation. These events were rescued by BAPTA-AM (Figure 4C). The transport of Cyt c from the mitochondria to the cytoplasm is mainly carried out by opening of the mPTP. Our data showed that BAPTA-AM inhibited HG or AGE-induced mPTP opening and further reduced the release of cytoplasmic Cyt c (Figure 4D). Mitochondrial Ca^2+^-induced elevation of mitochondrial ROS and a decrease in mitochondrial membrane potential induced mitochondrial dysfunction and promoted mPTP opening, which further induced cytoplasmic Cyt c/caspase-3-mediated apoptosis. Together, our findings suggested that mitochondrial Ca^2+^ overload induces mitochondrial dysfunction and apoptosis in RMECs.

### 3.4. ERS Triggered ER–Mitochondrial Coupling and Transferred Ca^2+^ into the Mitochondria

ERS and mitochondrial dysfunction in RMECs are the most evident pathological phenomena in DR [30]. However, the mechanisms underlying ERS and mitochondrial dysfunction in RMECs have not been elucidated. Ca^2+^ imbalance occurs when the ER is under stress due to alteration of Ca^2+^ uptake/release. To define whether ERS increases the ER-mitochondrial coupling, transfers Ca^2+^ into mitochondria, and promotes mitochondrial dysfunction, we used tunicamycin (TUN), a common ERS inducer. Interestingly, the results showed enhanced ER–mitochondrial coupling under TUN incubation (Figure 5A). Co-IP analysis indicated that there is enhancement of a combination of GRP75 with IP3R1 or VDAC1 in TUN-induced RMECs (Figure 5B). In addition, knockdown of GRP75 significantly rescued the upregulation of TUN-induced ER–mitochondrial coupling (Figure 5A). Furthermore, activation of ERS increased the transport of Ca^2+^ from the ER to the mitochondria (Figure 5C). These results indicated that ERS triggers ER–mitochondrial coupling through the IP3R1–GRP75–VDAC1 axis and transfer Ca^2+^ into the mitochondria.

ERS leads to production of mitochondrial ROS (Figure 5D), causing a decrease in mitochondrial membrane potential and subsequent opening of the mPTP (Figure 5E,F). The release of cytoplasmic Cyt c was confirmed by Western blot analysis (Figure 5G). We next evaluated the presence of apoptosis in TUN-induced RMECs. As shown in Figure 5H,I, there was upregulation of the expression of pro-apoptotic proteins (Bax and cleaved caspase-3), while the anti-apoptotic proteins (Bcl-2 and Bcl-xl) were downregulated following treatment with TUN. Moreover, the activation of ERS led to reduction in cell survival rate (Figure 5J,K). BAPTA-AM was shown to reverse the above phenomenon (Figure 5D,K), which further indicated that overload mitochondria Ca^2+^ induced mitochondrial dysfunction and apoptosis under ERS. Overall, these results demonstrate that ERS increases ER–mitochondrial coupling and transfers Ca^2+^ into the mitochondria and overload mitochondria Ca^2+^, leading to mitochondrial dysfunction and apoptosis.

### 3.5. Inhibition of ERS Ameliorated Retinal Dysfunction in Streptozotocin (STZ)-Induced DR Rats

The data demonstrated that ERS increased ER–mitochondrial coupling, leading to mitochondrial Ca^2+^ overload, which induces mitochondrial dysfunction and apoptosis of RMECs. To further confirm the roles of ERS in DR, we performed an in vivo experiment using 4-phenylbutyric acid (4-PBA), an inhibitor of ERS. As shown in Figure 6A, administration of 4-PBA (500 mg/kg, oral gavage, once a day) was initiated 2 weeks after STZ treatment. The data showed that 4-PBA significantly reversed STZ-induced retinal vascular leakage (Figure 6B). Moreover, retinal trypsin digestion showed STZ-induced capillary degeneration and pericyte loss, which were rescued by 4-PBA administration (Figure 6C).

The intraretinal microvascular abnormalities (IRMA), such as retinal vascular leakage and acellular capillaries, are classic hallmarks of DR. Retinal structural morphology was determined by HE staining and showed that 4-PBA ameliorated STZ-induced retinal thickness loss (Figure 6D). Vascular endothelial growth factor (VEGF) is a prominent vasopermeability factor that significantly contributes to DR. Brn3a is a marker of retinal ganglion cell (RGC) viability, and RGC degeneration is a hallmark of DR. Our results indicated that 4-PBA rescued STZ-induced upregulation of VEGF and downregulation of Brn3a, and further ameliorated DR (Figure 6E). In addition, rhodopsin, a compound of retinoids and optins, was suppressed in the DR rat model, which was reversed by the administration of 4-PBA (Figure 6F). In addition, our results showed that microglial activation and Müller cell gliosis were rescued by 4-PBA (Appendix A). These findings showed that inhibition of ERS ameliorates retinal dysfunction in STZ-induced DR rats. However, the specific mechanisms still need further clarification.

In addition, the STZ group showed an upregulation of CHOP, a marker of ERS, which was rescued by 4-PBA (Figure 7A). Thus, ERS is closely associated with DR. On the other hand, ERS and oxidative stress are closely related. Crosstalk between the ER and mitochondria increases oxidative stress, which can cause ERS. Oxidative stress in the retina was measured by 3-nitrotyrosine (3-NT), 4-hydroxynonenal (4-HNE), and 8-hydroxy-2′-eoxyguanosine (8-OHDG), which are mainly upregulated in DR. Our results showed that 4-PBA inhibited an STZ-induced increase in 3-NT, 4-HNE, and 8-OHDG (Figure 7B). ERS and oxidative stress generally induces apoptosis. TUNEL assay, a classical method of assessing apoptosis, showed that 4-PBA significantly ameliorates STZ-induced apoptosis in DR rats (Figure 7C). 4-PBA further reduced the levels of pro-apoptotic protein Bax and promoted the anti-apoptotic proteins Bcl-2 and Bcl-xL (Figure 7D), as well as inhibition of caspase-3-induced apoptosis (Figure 7E). These findings demonstrate that inhibition of ERS ameliorates retinal dysfunction and slows DR progression.

## 4. Discussion

The pathogenesis of DR, in which retinal vascular endothelial cell (RMEC) dysfunction plays a crucial role, is a complex process. However, the mechanisms involved in RMEC dysfunction have not been fully established. Various studies have evaluated the roles of ERS or mitochondrial dysfunction in RMECs during DR progression; however, the molecular mechanisms involved in interactions between the two organelles are yet to be conclusively determined. We found that ER–mitochondrial coupling was increased under the DR state and was associated with the upregulated IP3R1–GRP75–VDAC1 axis. Mitochondrial Ca^2+^ overload was induced by persistent Ca^2+^ transport from ER to the mitochondria, leading to mitochondrial dysfunction and RMECs damage. The main mechanisms are shown in Figure 8.

AGEs, H_2_O_2_, and HG are the frequently used agents for in vitro induction of diabetic-related complications [31,32]. The situation is more complex with exposure to HG, which closely resembles the human condition. AGEs, which are closely associated with DR, are products of excess glucose and protein formation in the body [33]. Therefore, HG and AGEs were used to simulate pathological changes in DR in vitro. Information communication between organelles is a universal physiological phenomenon. We established that HG and AGEs enhance ER–mitochondrial coupling in RMECs. Under physiological conditions, ER–mitochondrial coupling regulates Ca^2+^ and lipid transport between organelles, as well as the maintenance of mitochondrial morphology, dynamics, and energy metabolism [34,35]. However, in pathological conditions, increased or decreased ER–mitochondrial coupling induces various pathological changes, such as mitochondrial Ca^2+^ overload, mitochondrial dysfunction, lipid accumulation, and oxidative stress [36]. Therefore, studies have aimed at evaluating the roles of ER–mitochondrial coupling in disease progression.

Mitochondria-associated membranes (MAMs) are synaptic-like subcellular structures formed between the ER membrane and the mitochondria. As sites for bidirectional communication between ER and the mitochondria, MAMs play an essential role in regulating basic cell physiological activities [37]. Furthermore, MAMs are a particular class of cellular structures with flexible and plastic features and are regulated by various regulatory factors [18]. MAM-associated Ca^2+^ homeostasis is a research focus in metabolism-related diseases [38,39]. Ca^2+^ is an essential intracellular second messenger, regulating multiple pathophysiological processes in cells [40]. Intracellular Ca^2+^ homeostasis depends on coordination between enzymes, ion pumps, and Ca^2+^ channels, as well as storage of ER, mitochondria, and lysosomes [41,42]. The mitochondria rapidly detect and respond to changes in cytosolic Ca^2+^, which is an important buffer for regulating Ca^2+^ homeostasis [39]. Mitochondrial Ca^2+^ uptake is primarily regulated by VDAC, which anchors VDAC to Ca^2+^ release channels in ER/sarcoplasmic reticulum (SR) via scaffolding proteins [43]. IP3R and ryanodine receptor (RyR) are the main Ca^2+^ release channels on the ER, while GRP75 is a bridge between IP3R and VDAC [42]. Therefore, the IP3R1/GRP75/VDAC1 axis is a key pathway in the regulation of mitochondrial Ca^2+^ homeostasis on MAMs. In skeletal muscles of insulin resistance mice models, increased MAM formation mediated Ca^2+^ transport to the mitochondria via the IP3R1–GRP75–VDAC1 complex, resulting in mitochondrial Ca^2+^ overload and dysfunction [44]. In this study, we found increased MAM formation accompanied by the upregulated IP3R1/GRP75/VDAC1 axis and mitochondrial Ca^2+^ overload under DR conditions. However, GRP75 knockdown significantly reduced mitochondrial Ca^2+^ overload and dysfunction. Therefore, GRP75 may be a bridge protein linking ER–mitochondrial coupling and mitochondrial Ca^2+^. As a member of the HSP70 protein family, GRP75 is involved in formation of MAMs and participates in cellular endocytosis, proliferation, and tumorigenesis [45,46]. Apolipoprotein-E4-induced ERS in neurons upregulates GRP75 expressions, accelerates MAM formation, and impairs neuronal mitochondrial functions through mitochondrial Ca^2+^ overload [23]. In palmitate-induced pancreatic β cells, GRP75 overexpression promoted MAM formation and caused mitochondrial dysfunction [22]. However, after axon dissection in primary neurons, GRP75 overexpression enhanced the tricarboxylic acid cycle and increased ATP production by upregulating mitochondrial Ca^2+^, accelerating the regrowth of damaged axons [47]. Under different disease progression states, the ER–mitochondrial coupling alterations play different roles. Therefore, the roles of increased ER–mitochondrial coupling in RMECs under DR states should be investigated.

Hyperglycemia induces mitochondrial Ca^2+^ overload in RMECs [29]; however, the underlying mechanisms have yet to be fully elucidated. ER, as the principal site of Ca^2+^ storage, plays important roles in Ca^2+^ homeostasis. Increased ER–mitochondrial coupling is closely associated with mitochondrial Ca^2+^ overload. Various studies are evaluating the physiological significance of mitochondrial Ca^2+^ homeostasis [48,49]. Mitochondrial Ca^2+^ imbalance is involved in multiple disease processes, including diabetes, neurodegeneration, heart failure, and cancer [50,51,52]. However, the role of mitochondrial Ca^2+^ homeostasis in DR requires further studies. Mitochondria, the central site of energy metabolism, is closely correlated with Ca^2+^ homeostasis. Under physiological conditions, key enzymes of the tricarboxylic acid cycle are activated by Ca^2+^, thereby increasing ATP production to maintain higher energy requirements. However, under pathophysiological conditions, mitochondrial Ca^2+^ overload promotes mitochondrial ROS levels, mitochondrial membrane potential disruption, and mPTP opening in cells [53]. ER–mitochondrial coupling impairs mitochondrial membrane potential and increases mitochondrial Ca^2+^ in pancreatic β-cells [22]. We established that BAPT-AM (Ca^2+^ chelator) reverses hyperglycemia induced-mitochondrial dysfunction and apoptosis via the mPTP-mediated Cyt c/caspase-3 pathway, suggesting that mitochondrial Ca^2+^ overload accelerates the DR process via mitochondrial dysfunction.

Dysfunction of the ER and/or the mitochondria plays a significant role in DR development [30,54]. The mechanisms of ER–mitochondrial coupling are still unclear. Disruption of ER homeostasis results in accumulation of unfolded and misfolded proteins in the ER, leading to ERS. ERS may cause cell death by inducing Ca^2+^ overload, inhibiting protein synthesis and promoting apoptosis [55]. Khaled Elmasry et al. found that upregulated Ca^2+^ levels in the retina promotes 12/15-lipoxygenase-induced ERS and microvascular dysfunction by activating the ERS/NADPH oxidase/VEGFR2 signaling pathway [29]. In this study, TUN (an inducer of ERS) increased mitochondrial and ER coupling in RMECs, accompanied by mitochondrial Ca^2+^ overload, mitochondrial dysfunction, and apoptosis. These effects were associated with upregulated IP3R1–GRP75–VDAC1 axis.

This study has some limitations. First, retinal mitochondrial Ca^2+^ levels cannot be detected in vivo. Second, GRP75 knockout mice were not used. However, despite these limitations, we found that DR promotes MAM formation via the IP3R1–GRP75–VDAC1 axis, facilitates Ca^2+^ transport from the ER to the mitochondria, and contributes to mitochondrial Ca^2+^ overload, leading to mitochondrial dysfunction and endothelial cell apoptosis.

## 5. Conclusions

In this study, our findings imply that Ca^2+^ homeostasis plays an important role in DR. ER–mitochondrial coupling was increased under the DR state and was associated with the upregulated IP3R1–GRP75–VDAC1 axis. Mitochondrial Ca^2+^ overload was induced by persistent Ca^2+^ transport from ER to the mitochondria, leading to mitochondrial dysfunction and RMEC damage. 4-PBA, an ERS inhibitor, reversed retinal dysfunction and apoptosis in DR rat models. The results elucidate the underlying mechanisms of ER–mitochondria coupling in DR and showed that inhibited ER–mitochondrial coupling, including inhibition of GRP75 or Ca^2+^ overload, may be used as a potential therapeutic target for DR.

## Figures and Tables

**Figure 1 biomolecules-12-01778-f001:**
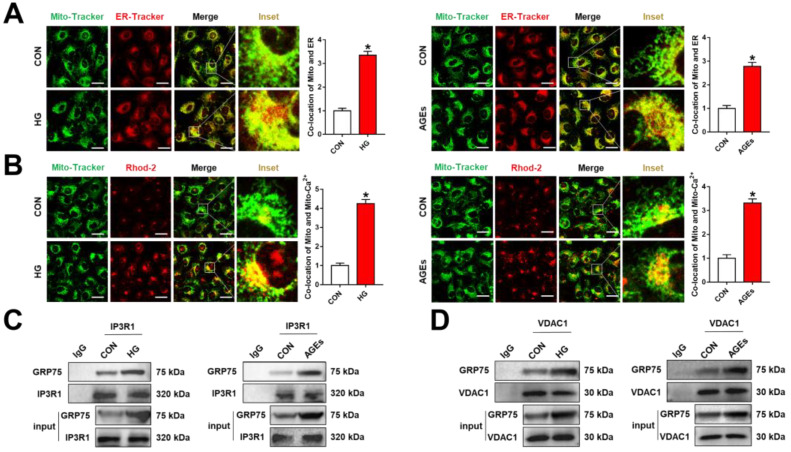
ER–mitochondrial coupling via the IP3R1–GRP75–VDAC1 axis in HG or AGE-induced RMECs. RMECs were treated with HG (33 mM) or AGEs (200 μg/mL) for 48 h. (**A**,**B**) Mito-Tracker Green (200 nM, 30 min), ER-Tracker Red (1 μM, 30 min), and Rhod-2 AM (2 μM, 30 min) were used as mitochondrial, ER, and mitochondrial Ca^2+^ markers, respectively. Co-localization of mitochondrial and ER, mitochondrial, and mitochondrial Ca^2+^ was inspected through confocal microscopy. Scale bars = 25 μm. (**C**,**D**) Co-immunoprecipitation analysis of GRP75 interactions with IP3R1 or VDAC1 in RMECs. Data are presented as means ± SD. * *p* < 0.05 vs. control group.

**Figure 2 biomolecules-12-01778-f002:**
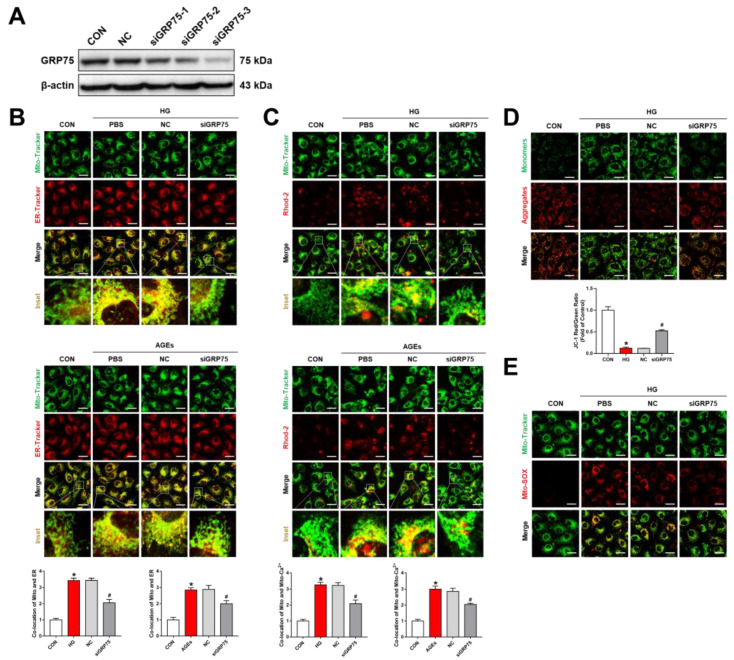
Inhibition of GRP75 rescued ER–mitochondria coupling and mitochondrial dysfunction in HG or AGE-induced RMECs. (**A**) RMECs were transfected with GRP75 siRNA with a 48 h period, and the transfection efficiency was estimated through Western blot. (**B**,**C**) RMECs were transfected with GRP75 siRNA for 48 h and stimulation with HG (33 mM) or AGEs (200 μg/mL) for 48 h. Mito-Tracker Green (200 nM, 30 min), ER-Tracker Red (1 μM, 30 min), and Rhod-2 AM (2 μM, 30 min) were used as mitochondrial, ER, and mitochondrial Ca^2+^ markers, respectively. Scale bars = 25 μm. (**D**) Mitochondrial membrane potential assessment was performed by JC-1 staining. The ratios of red fluorescence (JC-1 aggregates) and green fluorescence (JC-1 monomers) were quantified. Scale bars = 25 μm. (**E**) MitoSOX™ Red (5 μM, 10 min) and Mito-Tracker Green (200 nM, 30 min) were used to detect mitochondrial ROS co-localization with mitochondria. Scale bars = 25 μm. Results are displayed as means ± SD. * *p* < 0.05 vs. control group, ^#^
*p* < 0.05 vs. HG or AGEs group.

**Figure 3 biomolecules-12-01778-f003:**
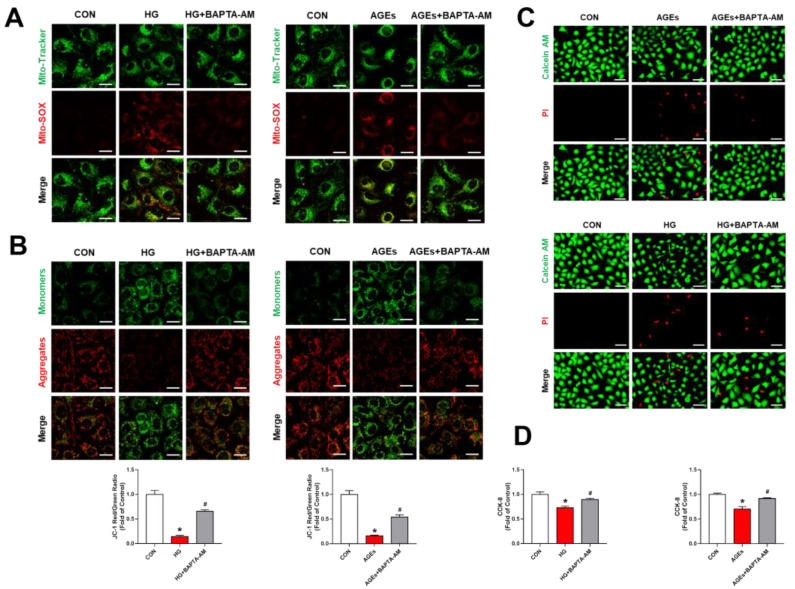
Mitochondria Ca^2+^ induced mitochondrial dysfunction and damage in RMECs. RMECs were preconditioned with BAPTA-AM (10 μM) for 2 h, supplemented by stimulation with HG (33 mM) or AGEs (200 μg/mL) for 48 h. (**A**) MitoSOX™ Red (5 μM, 10 min) and Mito-Tracker Green (200 nM, 30 min) were used to co-localize mitochondrial ROS and mitochondria. Scale bars = 25 μm. (**B**) Mitochondrial membrane potential was assessed by JC-1 staining. Scale bars = 25 μm. (**C**) Live cells (green, Calcein-AM, 2 μM, 30 min) and dead cells (red, PI, 5 μM, 30 min) were detected by a Calcein-AM/PI double staining kit. Scale bars = 25 μm. (**D**) BAPTA-AM rescued HG or AGE-induced cell damage, as determined by the CCK-8 assay. Data are presented as means ± SD. * *p* < 0.05 vs. control group, ^#^ *p* < 0.05 vs. HG or AGEs group.

**Figure 4 biomolecules-12-01778-f004:**
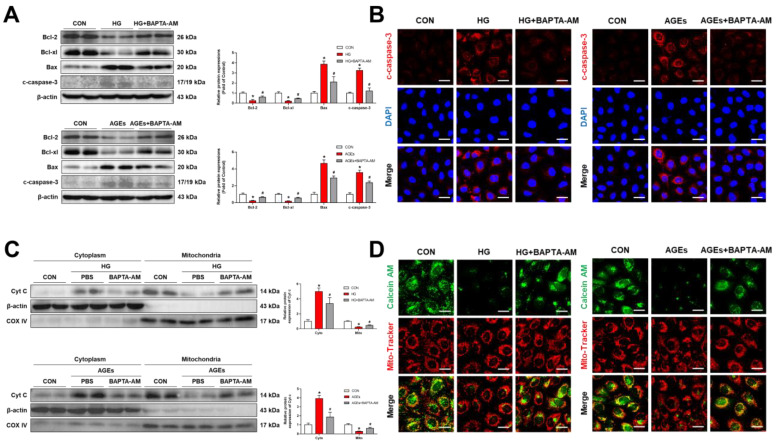
Mitochondrial Ca^2+^ induced RMECs apoptosis. RMECs were pretreated with BAPTA-AM (10 μM) for 2 h, then stimulated with HG (33 mM) or AGEs (200 μg/mL) for 48 h. (**A**) Apoptosis-related proteins (Bcl-2, Bcl-xl, Bax, and cleaved caspase-3) were assessed through Western blotting. (**B**) Cleaved caspase-3 was performed in a representative immunofluorescence assay. Scale bars = 25 μm. (**C**) Cyt c expressions in mitochondrial and cytosolic fractions were measured through Western blotting. (**D**) To detect mitochondrial permeability transition pores (mPTPs), RMECs were loaded with a Calcein-AM dye (2 μM) containing CoCl_2_ (250 μM) for 45 min. Mito-Tracker Red (200 nM, 30 min) was used to co-stain the mitochondria. Scale bars = 25 μm. Results are presented as means ± SD. * *p* < 0.05 vs. control group, ^#^ *p* < 0.05 vs. HG or AGEs group.

**Figure 5 biomolecules-12-01778-f005:**
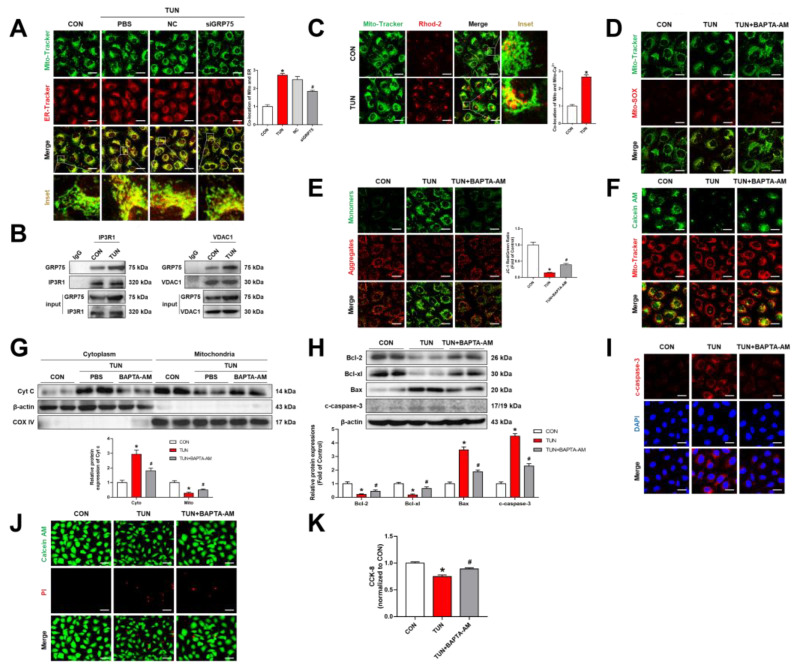
ERS induced ER–mitochondrial coupling, leading to mitochondrial dysfunction and cell apoptosis. RMECs were transfected with GRP75 siRNA for 48 h and incubated with TUN (5 μg/mL) for 24 h. (**A**) Mito-Tracker Green (200 nM, 30 min) and ER-Tracker Red (1 μM, 30 min) were used to determine co-localizations of the mitochondria and ER. (**B**) Co-immunoprecipitation analysis of GRP75 interactions with IP3R1 or VDAC1 in RMECs. (**C**,**D**) Mitochondrial Ca^2+^ and mitochondrial ROS were detected by Rhod-2 AM (2 μM, 30 min) and MitoSOX™ Red (5 μM, 10 min). Mito-Tracker Green (200 nM, 30 min) was used to co-localize the mitochondria. (**E**) Mitochondrial membrane potential was assessed by JC-1 staining. (**F**) Mitochondrial permeability transition pores (mPTPs) were detected using a Calcein-AM dye (2 μM) containing CoCl_2_ (250 μM) for 45 min. (**G**) Cyt c expressions in mitochondrial and cytosolic fractions were detected by Western blotting. (**H**) Apoptosis-related proteins (Bcl-2, Bcl-xl, Bax, and cleaved caspase-3) were evaluated by Western blotting. (**I**) cleaved caspase-3 was performed in a representative immunofluorescence assay. (**J**) Live cells (green, Calcein-AM, 2 μM, 30 min) and dead cells (red, PI, 5 μM, 30 min) were probed through the use of a Calcein-AM/PI double staining kit. (**K**) BAPTA-AM rescued TUN-induced cell damage, as determined by the CCK-8 assay. Data are presented as means ± SD. * *p* < 0.05 vs. control group, ^#^ *p* < 0.05 vs. HG or AGEs group.

**Figure 6 biomolecules-12-01778-f006:**
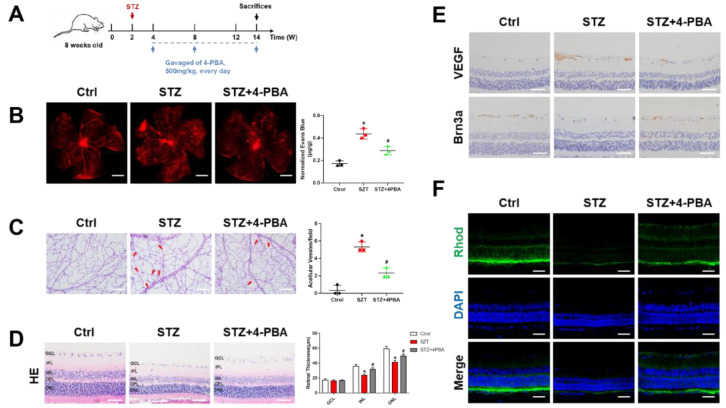
Inhibition of ERS ameliorated retinal dysfunction in DR rats. (**A**) Schematic presentation of in vivo procedures. (**B**) Rats with a tail vein injection of Evans blue dye (3%) for 2 h. Red fluorescent spots indicate retinal vascular leakage. (**C**) Representative images of acellular capillaries are shown by retinal trypsin digestion experiments and quantified by randomized regions. Scale bars = 50 μm. (**D**) Representative images of HE staining of rat retinas. The thickness of ONL, INL, and GCL layers of rats were assessed in the sections. Scale bars = 50 μm. (**E**) Immunohistochemistry analysis of VEGF and Brn3a in retinas. Scale bars = 50 μm. (**F**) Immunofluorescence staining for Rhod (green) and DAPI (blue) was performed in rat retinal sections. Scale bars = 50 μm. Data are presented as mean ± SD. * *p* < 0.05 vs. control group, ^#^ *p* < 0.05 vs. STZ group.

**Figure 7 biomolecules-12-01778-f007:**
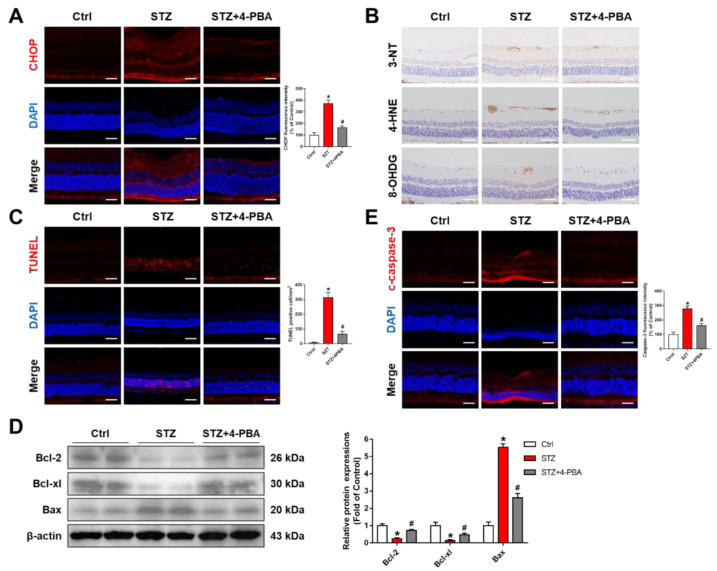
Inhibition of ERS ameliorated apoptosis in DR rats. (**A**) Immunofluorescence staining for CHOP (red) and DAPI (blue) was performed in rat retinal sections. Scale bars = 50 μm. (**B**) Immunohistochemistry analysis of 3-NT, 4-HNE, and 8-OHDG in retinas. Scale bars = 50 μm. (**C**) Apoptosis in retinal sections was appraised with the TUNEL assay. Scale bars = 50 μm. (**D**) Western blot analysis for protein expressions of apoptosis-related proteins (Bax, Bcl-2, and Bcl-xl). (**E**) Cleaved caspase-3 (red) and DAPI (blue) in retinal sections were subjected to immunofluorescence staining. Scale bars = 50 μm. Data are presented as mean ± SD. * *p* < 0.05 vs. control group, ^#^ *p* < 0.05 vs. STZ group.

**Figure 8 biomolecules-12-01778-f008:**
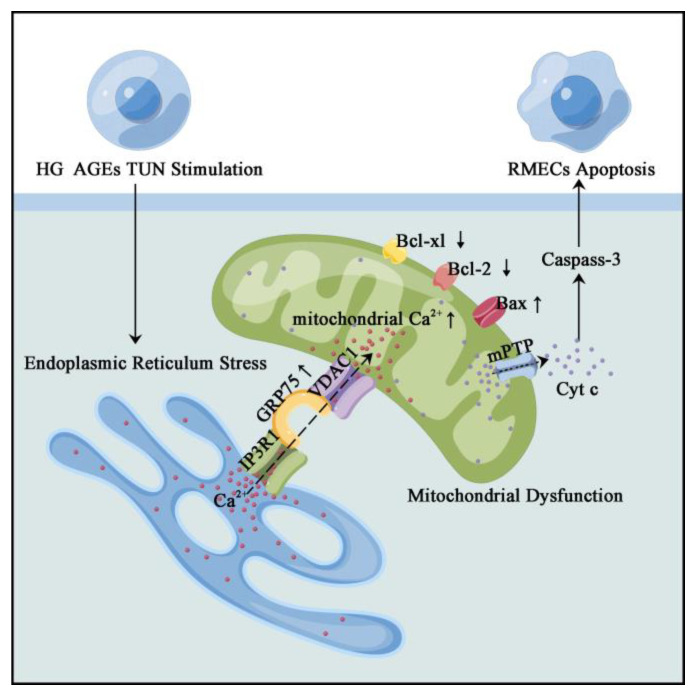
Possible mechanism of ER–mitochondria coupling and Ca^2+^-dependent endothelial cell apoptosis in DR. DR promotes ER–mitochondria coupling via the IP3R1–GRP75–VDAC1 axis and facilitates Ca^2+^ transport from the ER to the mitochondria, leading to mitochondrial dysfunction and RMECs apoptosis. Inhibited ER–mitochondrial coupling, including inhibition of GRP75 or Ca^2+^ overload, may be used as a potential therapeutic target for DR.

## Data Availability

Not applicable.

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
