# Peer review of "GRP75 Modulates Endoplasmic Reticulum–Mitochondria Coupling and Accelerates Ca2+-Dependent Endothelial Cell Apoptosis in Diabetic Retinopathy"

_biomolecules, 2022, doi:10.3390/biom12121778_

Round 1

Reviewer 1 Report

Li and colleagues provide a novel and interesting study regarding the role of ERS and ER/mitochondrial coupling in the pathogenesis of DR. The study is well designed and scientifically sound, with good experimental execution. Although the results are clearly presented and the conclusions are drawn carefully in full awareness of the limitations of the study, some points have to be mentioned:

Methods:

- please provide details on the in vitro stimulation (high glucose and AGE groups), why was the AGE modified BSA chosen for stimulation? Were the stimulations performed in otherwise serum-free media? How was the control group treated? Was there an osmotic control?

- please clarify the route of administration of 4-PBA in the method section

- The diabetes duration of 14 weeks was chosen quite long for Sprague Dawley rats, as acellular capillary formation starts as early as of 8 weeks diabetes duration (Xie et al., Diabetologia 2021 DOI: 10.1007/s00125-020-05299-x) -> Could a shorter time period have been more appropriate as the mechanism of the intervention targets early effects?

Results and Discussion:

- The in vivo results show that nearly every compartment of the retinal neurovascular unit benefitted from the treatment. However, as the diabetes duration was quite long, it is hard to determine which of these effects are secondary to a, possible, beneficial effect on the vasculature. Some experiment showing the primary effects on the vasculature in accordance with the extensive in vitro experiments would be of great value for this study. In addition, the glial compartment was left out entirely, additional data on microglial activation and Müller cell gliosis would complete the in vivo results.

Reviewer 2 Report

Here the authors present the evidence that activation of ERS in DR due to HG and AGE increase the release of calcium in mitochondria. This leads to the dysfunction of mitochondria and damage to RMECs. 

Comments for the Authors

1.              Line 25-26: “The above phenomenon can was also observed in tunicamycin (TUN, ER stress inducer)-treated cells”.  This line should be corrected. Please check the manuscript for more grammatical errors. 

2.              In the introduction, reference#2 is not the correct reference to describe DR complications. Several literature reviews describe DR complications which can be referenced here. 

3.              Line 40-41: “Since retinal microvascular endothelial cells (RMECs) line the inner wall of vessels, its dysfunction is an early event in DR due to long-term contact with glucose-rich blood”. Please provide a reference for the statement. 

4.              It is mentioned in the methods that SD rats were 8 weeks old when injected with STZ. After 2 weeks of STZ injection, the rats were treated with PBA from the 2nd week to the 14th week. Please check the timeline diagram in fig 6. A. As per the figure, the rats received PBA from 3rd week post-STZ injection. Can authors please clarify?

5.              The authors injected STZ as a single dose and 70% of rats developed hyperglycemia in this experiment. Usually, 200mg/dl is the hyperglycemia limit for rats, but here, the authors have a hyperglycemia limit of 300mg/dl. Can the authors please provide a reference for the dosage and the blood glucose level of 300mg/dl thresholds?

6.              Can the authors please provide information regarding, 

a.     how many cells were plated per experiment?

b.     Are the figures representative of one well?

c.     What is the sample size? N= # of wells (representing biological variants) or N=# technical replicates from all the pooled cells. 

7.              In all the figures with RMECs, why did the authors not use the original images provided in PowerPoint? The images seem to have been cropped to focus on the cells in the center of the original image. This seems to be true for every IHC image of cells and it is difficult to compare the original images with the manuscript image panel for some figures. Please check all the inset boxes for alignment, especially, fig1A HG and AGE control merge. 

8.              The authors mention that the IHC images in Fig 1 suggest an increase or decrease. This measure is however only qualitative. It would be better if the authors can calculate the exposure intensity of the biomarkers and provide more semi-quantitative evidence. 

9.              Fig 2 D- should the y-axis be “ratio” instead of “radio”? This seems to be present in every graph with the corresponding title throughout the manuscript.

10.           Line 325-326: “Western blot analysis showed that BAPTA-AM inhibited HG....”. May need to rewrite, as it conveys that the BAPTA-AM increased both pre-apoptotic and anti-apoptotic proteins. 

11.            

a.     Fig 6B is the immunofluorescent intensity of the Evan blue dye leakage normalized to the whole retinal area of the flat mount. Please explain the unit mg/mg for the y-axis. Does it represent the amount of dye injected or detected?

b.     Fig 6C and Fig 6D have the legends interchanged. Please correct. 

c.     Sometimes in cross-sections, the overall thickness can be difficult to detect with any artifacts in the processing of the retina. If authors could check the thickness of ONL, INL, and RGC layers, it would strengthen their data (Fig 6C). Also, if the authors could specify the location of all the sections, such as the mid-periphery or periphery of the retina, it would be helpful as the thickness of the retinal cross-section can vary at locations. 

d.     In fig 6D the red arrows in the panels are very difficult to see due to their size. Please correct. 

12.   

a.     Fig 7A-E. The IHC images provide qualitative analysis, authors should at least measure the intensity of IHC markers (CHOP and CASPASE-3) to conclude an increase or decrease in expression.  

b.     Fig 7C Tunnel assay- were the TUNEL-positive nuclei counted manually or automatically? Usually, quantitatively, the number of tunnel positive nuclei is presented as normalized to the area of the explant/cross-section/image.

Round 2

Reviewer 2 Report

Dear Authors,

Thank you for responding to the comments. 

I have no more comments for the manuscript.

Thanks.